# Resistance of Field-Isolated Porcine Epidemic Diarrhea Virus to Interferon and Neutralizing Antibody

**DOI:** 10.3390/vetsci9120690

**Published:** 2022-12-11

**Authors:** Jung-Eun Park, Hyun-Jin Shin

**Affiliations:** 1Laboratory of Veterinary Public Health, College of Veterinary Medicine, Chungnam National University, Daejeon 34134, Republic of Korea; 2Laboratory of Veterinary Infectious Diseases, College of Veterinary Medicine, Chungnam National University, Daejeon 34134, Republic of Korea

**Keywords:** porcine epidemic diarrhea virus, resistance, interferon, neutralizing antibody

## Abstract

**Simple Summary:**

Porcine epidemic diarrhea (PED) is a porcine enteric viral disease that is characterized by watery diarrhea, vomiting, and dehydration. PED is highly contagious and causes significant mortality, especially in suckling piglets. The G2b variant PEDV, which started to become prevalent in China in 2013, has higher pathogenicity and transmissibility than existing viruses and causes economic losses in many countries, including South Korea. The present study shows that the G2b variant PEDV has a relatively high resistance to interferons and neutralizing antibodies. The present study has important implications for understanding the occurrence of variant PEDV and its pathogenesis.

**Abstract:**

Variant porcine epidemic diarrhea virus (PEDV), belonging to the genogroup G2b, has higher pathogenicity and mortality than classical PEDV, belonging to the genogroup G1a. To understand the pathogenesis of the G2b PEDV, we examined the resistance of the G2b PEDV to interferon (IFN) and neutralizing antibodies, which are important for controlling PEDV infection. We found that the G2b PEDV showed higher resistance to IFN than G1a PEDV. The G1a PEDV could replicate in IFN-deficient Vero cells, but not in IFN-releasing porcine alveolar macrophages, whereas the G2b PEDV showed similar infectivity in both types of cells. We also found that G2b PEDV was not effectively blocked by neutralizing antibodies, unlike G1a PEDV, suggesting differences in the antigenicity of the two strains. These results provide an understanding of the occurrence of variant PEDV and its pathogenesis.

## 1. Introduction

Porcine epidemic diarrhea (PED) is a viral enteric disease in pigs caused by porcine epidemic diarrhea virus (PEDV). It occurs in pigs of all ages, from neonates to sows or boars. The incubation period is 26~36 h for newborn piglets and 2 days for finishing pigs. The incidence rate is close to 100% in newborn pigs and breeding pigs, but is as low as 10–90% in adult and breeding pigs. The symptoms are more severe at a young age; newborn pigs show anorexia and vomiting immediately after diarrhea, and suddenly develop watery diarrhea [1,2]. This leads to severe dehydration, and death occurs after 3 to 4 days of diarrhea. The mortality rate varies according to age; for piglets under 1 week of age, the average is 50%, but in severe cases, it rises to 90%.

PED was first identified in Europe in the 1970s [3], and has become geographically restricted in Europe and Asia over the past 30 years [4,5,6]. PED is not on the list of diseases reported to the World Animal Health Organization. Moreover, in countries where the disease is endemic, the impact is considered low. However, the emergence and outbreak of variant PEDV in naïve populations can result in significant losses and economic impacts. Since 2013, the spread of new variants of PEDV to North America caused huge economic losses to the swine industry [1,2]. 

PEDV, a member of the family Coronaviridae, genus *Alphacoronavirus*, is an enveloped single-stranded positive-sense RNA virus [3]. The viral genome is approximately 28 kb long and encodes at least seven open reading frames (ORFs): ORF1a, ORF1b, spike (S), ORF3, envelope, membrane, and nucleocapsid [3,4]. According to the S gene sequences, PEDV is genetically divided into two groups: genogroup 1 (classical or recombinant, low pathogenic) and genogroup 2 (field epidemic or pandemic, highly pathogenic) [5,6,7,8,9]. Each group is further divided into two subgroups: G1a (vaccine strains), G1b (new variants), G2a (past epidemic strains), and G2b (current dominant epidemic strains).

The host cell’s innate immune response prevents the spread of the infection, and initiates an adaptive immune response that clears the virus from the host to defend against virus infection. In particular, the type I/III interferon (IFN) plays a pivotal role in generating antiviral responses in immune and epithelial cells [10,11]. Humoral immune responses induced by vaccines provide protective immunity in neonatal suckling piglets from PEDV [12]. Coronaviruses, including PEDV, have developed a series of sophisticated mechanisms to evade the host’s antiviral innate immune response and vaccine-derived neutralizing antibodies.

The G2b PEDVs have emerged in several countries, including those that conduct annual PEDV vaccinations, and have induced enormous economic losses. We hypothesized that the G2b PEDVs are resistant to factors that are known to be important for PEDV control. To test this hypothesis, we compared the susceptibility against IFN and neutralizing antibodies using the G1a CV777-lineage classical and the G2b strains of PEDV.

## 2. Materials and Methods

### 2.1. Cells

African green monkey kidney (Vero, KCLB 10081) cells were maintained in Dulbecco’s Modified Eagles Medium (DMEM, Hyclone, Logan, UT, USA) complemented with 10% (*v*/*v*) fetal bovine serum (FBS, Hyclone), 100 U/mL penicillin G (Hyclone), and 100 μg/mL streptomycin (Hyclone). Intestinal porcine epithelial cells from jejunum (IPEC-J2) cells were kindly obtained from Dr. Hyun Jang (Libentech Co., Ltd., Daejeon, Republic of Korea) and maintained in DMEM supplemented with 10% FBS, 1X Insulin-Transferrin-Selenium-G (Gibco, Billings, MT, USA), 100 U/mL penicillin G, and 100 μg/mL streptomycin. Porcine alveolar macrophage 3D4/21 (ATCC CRL-2843™) cells were maintained in the Roswell Park Memorial Institute 1640 medium (RPMI-1640, Gibco), complemented with 10% FBS, 100 U/mL penicillin G, and 100 μg/mL streptomycin. The cells were maintained at 37 °C under 5% CO_2_.

### 2.2. Virus Infection

PEDV SM98 (cell culture-adapted vaccine strain, G1a, >70 passages) and PED-CUP-B2014 (field-isolated strain, G2b, passage 15~20) were propagated in Vero cells as previously described [13,14]. The cells were pre-washed twice with DMEM containing antibiotics, and then were incubated with PEDV for 1 h at 37 °C at an MOI of 0.1. After 1 h, the inocula were aspirated and the cells were incubated in DMEM with or without 10 µg/mL trypsin (Sigma, Kawasaki, Japan). At the indicated time points, cells were freeze-thawed three times and centrifuged at 1000× *g* for 10 min. Supernatants were harvested for virus titration.

For IFN treatment, IPEC-J2 cells were treated with recombinant porcine IFN-β (Abcam, Cambridge, UK) for 24 h. The cells were infected with either SM98 or PED-CUP-B2014 at an MOI of 1. After 1 h, inocula were aspirated and the cells were incubated in DMEM containing antibiotics. At 24 h post-infection (hpi), cell-free supernatants were harvested for virus titration.

### 2.3. Virus Titration

PEDV titers were determined using the Median Tissue Culture Infectious Dose (TCID_50_) assay. Vero cells were pre-washed twice with DMEM containing antibiotics, and were then infected with 10-fold serially diluted virus for 1 h at 37 °C. After 1 h, the inocula were aspirated and the cells were kept in DMEM containing 5 μg/mL trypsin. After 5 days, the cells were stained with 1% crystal violet staining solution (Sigma).

### 2.4. Antiserum Production

Animal experiments were performed following the protocol permitted by the Institutional Animal Care and Use Committee of Chungnam National University (CNU-01184). Eight-week-old female BALB/c mice were divided into three groups of six mice each. Viruses were concentrated by centrifugation at 100,000× *g* for 2 h at 4 °C and suspended in phosphate-buffered saline (PBS). The mice were injected intraperitoneally twice at 2-week intervals with SM98, PED-CUP-B2014, or PBS. At 28 days after initial immunization, mice were euthanized and blood samples were collected via cardiac puncture. 

### 2.5. Enzyme Linked Immunosorbent Assay (ELISA)

Microplates were coated with the indicated PEDV (10^4^ TCID_50_/0.1 mL) at 4 °C overnight. The plates were washed thrice with 400 μL of PBS-T (PBS containing 0.05% Tween 20) and blocked with 100 μL blocking solution (PBS-T containing 2% bovine serum albumin) for 2 h at 25 °C. Plates were then incubated with serum samples diluted to 1:100 in blocking solution for 2 h at 37 °C. After washing the plates, the plates were incubated with 100 μL of horseradish peroxidase-conjugated goat anti-mouse IgG for 1 h at 37 °C. After washing the plates, the plates were incubated with 100 μL of 3,3,5,5-tetramethylbenzidine solution for 30 min at 25 °C in the dark, and the reaction was stopped by adding 100 μL of 2N sulfuric acid. Optical density (OD) was measured at 450 nm using VICTOR Nivo Multimode Microplate Reader (PerkinElmer, Waltham, MA, USA).

### 2.6. Serum Neutralization (SN) Assay

SN titers were measured as previously described [13]. Serum samples were heat-inactivated at 56 °C for 30 min. Serum samples were then 2-fold serially diluted in DMEM containing antibiotics, and then mixed with PEDV virus (200 TCID_50_/0.05 mL). Mixtures were incubated at 37 °C for 90 min. Vero cells plated in 96-well plate were washed twice with DMEM containing antibiotics, and infected with 100 ul of the serum–virus mixture at 37 °C for 1 h. After 1 h, inocula were aspirated and cells were incubated in DMEM containing 5 ug/mL trypsin and antibiotics for 48 h at 37 °C. The medium was then removed, and the cells were stained with 1% crystal violet staining solution for 1 h at 25 °C. The SN titer was calculated as the reciprocal of the highest dilution of serum that inhibited the virus-specific cytopathic effect in all replicating cells. 

### 2.7. Statistical Analysis

All experiments were independently repeated at least twice. The sample size was decided based on previous experiences, and the number of samples for each experiment is indicated in figure legends. Data are presented as the mean ± standard deviation (SD). Statistical significance was determined using the Holm–Sidak multiple Student’s *t*-test. A *p* value of <0.05 was considered statistically significant. All graphs and statistical analyses were performed using GraphPad Prism software version 9.4.1.

## 3. Results

### 3.1. Infectivity of PEDV Strains SM98 and PED-CUP-B2014 in Various Cells 

The virus growth of SM98 and PED-CUP-B2014 strains was compared in Vero, IPEC-J2, and 3D4/21 cells. As Vero cells are IFN-deficient, they do not secrete IFNs when infected by viruses, but still have the IFN-alpha/beta receptor. Thus, they respond normally when recombinant IFN is added to culture media [15]. In Vero cells, SM98 proliferated well regardless of trypsin, but a greater amount of progeny viruses was produced in the presence of trypsin (Figure 1A). In contrast, virus growth in the PED-CUP-B2014 strain was only observed in the presence of trypsin. When comparing growth rates in the trypsin condition, the PED-CUP-B2014 strain showed a titer approximately 150 times lower at 24 hpi, but showed the same titer as that of the SM98 strain at 48 hpi (Figure 1A). In the IPEC-J2 cells, the SM98 strain proliferated only in the presence of trypsin, whereas PED-CUP-B2014 proliferated regardless of trypsin (Figure 1B). In the 3D4/21 cells, the SM98 strain did not proliferate in the absence of trypsin (Figure 1C). Even in the presence of trypsin, the amount of progeny viruses increased to 2.8 TCID_50_ at 24 hpi and then decreased, and no progeny viruses were detected at 48 hpi (Figure 1C). The proliferation of PED-CUP-B2014 was only observed in the presence of trypsin, and the virus titer reached 5.5 TCID_50_ and continued until 24 hpi (Figure 1C). The data indicated that PED-CUP-B2014, but not SM98, can grow in IFN-releasing cells.

### 3.2. IFN Susceptibility of PEDV Strains SM98 and PED-CUP-B2014

Based on the results shown in Figure 1, we hypothesized that PED-CUP-B2014 was resistant to IFN and proliferated well in 3D4/21 cells, which is very different from the SM98 strain. To confirm this hypothesis, we compared the infectivity of both viruses after the cells were treated with porcine IFN-β. As shown in Figure 2, the proliferation of the PED-CUP-B2014 strain was observed regardless of the IFN-β concentration, whereas the proliferation of the SM98 strain was inhibited by IFN-β in a dose-dependent manner. These results demonstrate that the PED-CUP-B2014 strain is more strongly resistant to IFN-β.

### 3.3. Susceptibility of PEDV Strains SM98 and PED-CUP-B2014 to Neutralizing Antibody

PED vaccination in sows has shown that neutralizing antibodies provided through colostrum play an important role in the prevention of PEDV in piglets [16]. The neutralizing activity of the antibodies induced by SM98 or PED-CUP-B2014 against each virus was evaluated. BALB/C mice were immunized with SM98 or PED-CUP-B2014, and serum samples were collected at 28 days post-immunization. The IgG levels against SM98 as the coating antigen were 1.093 ± 0.047 in SM98 antisera and 0.795 ± 0.103 in PED-CUP-B2014 antisera, respectively (Figure 3A). The IgG levels against PED-CUP-B2014 as the coating antigen were 1.570 ± 0.031 in SM98 antisera and 2.406 ± 0.077 in PED-CUP-B2014 antisera, respectively (Figure 3B). The SN titers against SM98 of both antisera were similar, with 204.8 ± 70.1 in SM98 antisera and 166.4 ± 85.9 in PED-CUP-B2014 (Figure 3C). The SN titers against PED-CUP-B2014 were six-fold higher in PED-CUP-B2014 (128.0 ± 78.4) than in SM98 antisera (21.6 ± 24.3) (Figure 3D). These results indicate that the PED-CUP-B2014 strain is resistant to antibodies against SM98 strains.

## 4. Discussion

The PEDV vaccine using G1a strain is extensively used in some Asian countries, including South Korea, China, Japan, and Thailand. Despite extensive vaccination, PEDV outbreaks remain a major concern in the pork industry [17,18,19]. In 2011, PEDV infection rates increased substantially in vaccinated swine herds [20,21]. The new variant of PEDV shows higher pathogenicity and transmissibility, and is genetically different from the previous PEDV strains [1,5,6]. The experiment described herein demonstrates that G2b PEDV is resistant to IFN- and vaccine strain-derived neutralizing antibodies. This suggests that this variant of PEDV confers increased pathogenicity in pigs.

Type I IFN is a potent cytokine which is critical in controlling viral infections and priming adaptive immune responses [22]. Type I IFNs bind their receptors on the cell surface and induce the expression of IFN-stimulated genes, which establish an antiviral state [23,24,25]. The capability to escape type I IFN is a key factor in virus virulence, survival, and successful infection. In this study, we found that PED-CUP-B2014 infected Vero, IPEC-J2 cells, and 3D4/21 cells (Figure 1). In contract, SM98 infected Vero and IPEC-J2 cells, but not 3D4/21 cells (Figure 1). In Vero cells, PED-CUP-B2014 replicated only in the presence of trypsin, because trypsin is indispensable in PEDV isolation [19]. Surprisingly, PED-CUP-B2014 replicated in IPEC-J2 cells regardless of trypsin treatment. We speculate that IPEC-J2 cells contain cellular proteases that may allow S cleavages, and further studies to address the host proteins are needed. We also found that PED-CUP-B2014 is more resistant to exogenous IFN-β treatment than SM98 (Figure 2). We speculate that increased resistance to IFN allowed PED-CUP-B2014 to grow in Vero as well as in alveolar macrophage-derived 3D4/21 cells (Figure 1). Although we did not test the pathogenicity of the viruses, several studies of antiviral treatments have shown that treatment of type I IFN inhibits viral propagation in cultured cells as well as viral replication in animal models [26,27,28]. Thus, we speculate that IFN resistance of field-isolated PEDV results in greater pathogenicity and transmissibility in pigs.

Most viruses have evolved diverse strategies to prevent the initiation of antiviral effectors in host cells, mostly by decreasing IFN production and impeding IFN signaling [29]. In coronaviruses, accessory proteins and some non-structural proteins (nsps) have been observed to play a role in host immune modulatory functions [30]. Nevertheless, the role of ORF3 in regulating host innate immunity remains as mysterious as its role in pathogenesis [31]. In contrast, nsp1, nsp3, and nsp5 were shown to be potent IFN antagonists [31,32,33]. In the present study, we have not studied which proteins in field-isolated PEDV play a role in IFN resistance. Further studies are needed to reveal the proteins participating in increased IFN resistance of field-isolated PEDV.

Antigenic differences of field-isolated PEDV have been suggested in previous studies [34,35]. It was also demonstrated that the current vaccines do not effectively protect field-isolated PEDV due to their antigenic differences [35]. Our data also support the hypothesis that the two strains are antigenically different. It is worth noting that field-isolated PEDV produced more IgG than the vaccine strain, suggesting that field-isolated PEDV is more immunogenic. Furthermore, antiserum against field-isolated viruses showed similar neutralizing activities to both field-isolated and vaccine strains. These data suggest that vaccines using field-isolated viruses are more effective. Several studies, including ours, have described the development and immune responses of vaccines with field-isolated strains [13,35,36]. 

One of the limitations of our study was the use of a limited number of virus strains and samples. Although vaccine strains show similar characteristics among themselves, field isolates have different mutations in their S proteins. Therefore, additional studies are needed to determine whether viruses with different S mutations have similar properties. Furthermore, the concentrations of recombinant IFN-β used in the study are much higher that the physiological levels of IFN-β. Therefore, it is necessary to validate whether PED-CUP-B2014 exhibits increased resistance to physiological levels of IFN-β. Finally, considering that the suitable host for PEDV is pigs, further studies using antisera from pigs infected with these viruses should be conducted.

## 5. Conclusions

PEDV is one of the most important swine viruses that has emerged or re-emerged, posing a significant threat to the global pork industry. In particular, the highly pathogenic PEDV strain, which began to spread in China in 2013, emerged in the United States and then spread to Asian countries such as Korea, Taiwan, and Japan almost simultaneously, causing PED epidemics nationwide. In the present study, we demonstrated that highly pathogenic PEDV strains are more resistant to IFNs and neutralizing antibodies com-pared to endemic or classical PEDV strains. Our results may provide a underlying mechanism for how highly pathogenic PEDV strains in the field emerge and spread in vaccinated pigs, and also suggest ways to control PEDV outbreaks of highly virulent strains.

## Figures and Tables

**Figure 1 vetsci-09-00690-f001:**
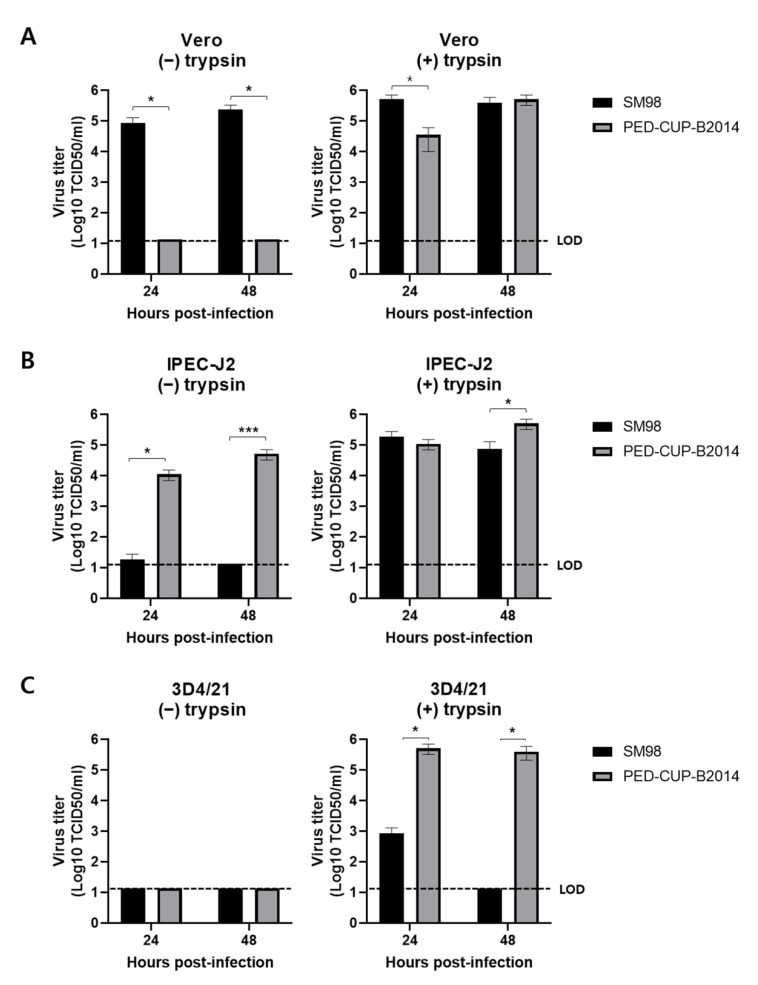
Infectivity of PEDV strains SM98 and PED-CUP-B2014 in various cells. Vero (**A**), IPEC-J2 (**B**), and 3D4/21 (**C**) cells were infected with SM98 or PED-CUP-B2014, with or without 10 µg/mL trypsin. At 24 or 48 h post-infection, virus titers were determined using TCID_50_ assay. Results are expressed as the mean (*n* = 3) ± SD. *, *p* < 0.05; ***, *p* < 0.0001. LOD, limit of detection.

**Figure 2 vetsci-09-00690-f002:**
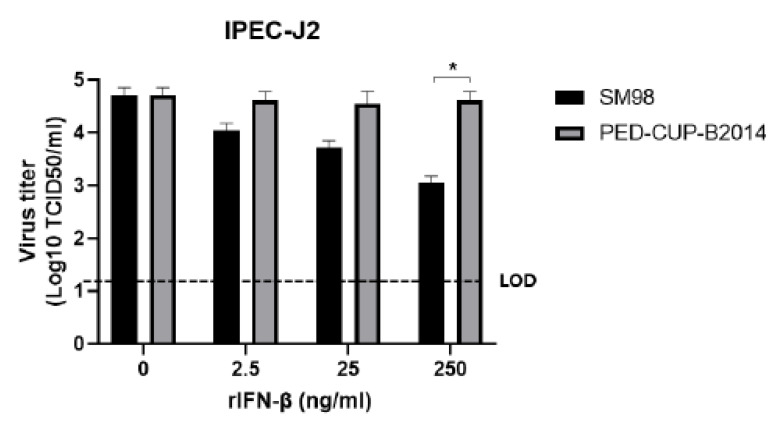
IFN susceptibility of PEDV strains SM98 and PED-CUP-B2014. IPEC-J2 cells were treated with the indicated concentrations of porcine recombinant IFN-β for 1 h, and then infected with SM98 or PED-CUP-B2014 for 1 h in the presence of 10 µg/mL trypsin and IFN- β. At 12 h post-infection, virus titers were determined using TCID_50_ assay. Results are expressed as the mean (*n* = 3) ± SD. *, *p* < 0.05. LOD, limit of detection.

**Figure 3 vetsci-09-00690-f003:**
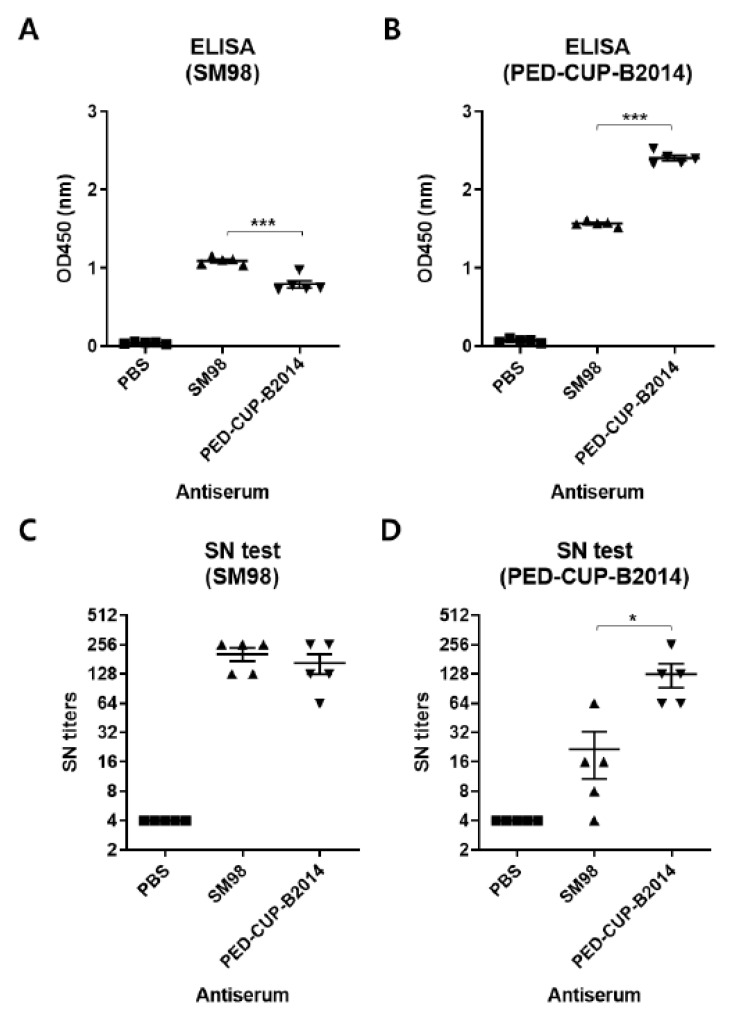
Susceptibility of PEDV strains SM98 and PED-CUP-B2014 to neutralizing antibodies. BALB/c mice were intraperitoneally immunized with SM98 or PED-CUP-B2014 twice, with a 2-week interval. Serum samples were collected 28 days post-immunization. (**A**,**B**) SM98- (**A**) or PED-CUP-B2014- (**B**) specific IgG levels were determined using ELISA. Results are expressed as the mean (*n* = 5) of OD450 ± SD values. (**C**,**D**) Neutralizing antibody titers against SM98 (**C**) and PED-CUP-B2014 (**D**) were measured using the serum neutralization test. Results are expressed as the mean (*n* = 5) of SN titers ± SD. *, *p* < 0.05, ***, *p* < 0.001.

## Data Availability

The entire datasets in the current study are available from first authors upon reasonable request.

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
