# Peer review of "Resistance of Field-Isolated Porcine Epidemic Diarrhea Virus to Interferon and Neutralizing Antibody"

_vetsci, 2022, doi:10.3390/vetsci9120690_

Round 1
Reviewer 1 Report (Previous Reviewer 2)
Most of the observations advanced in the previous review were considered by the authors.
However, the fact remains that the statistical part is very weak, as the number of samples, and if the authors wish to keep this part of the study in the manuscript, they must report it “as a limit of the study” in the discussion of the results.
Author Response
Thank you for your comments. Following your suggestion, we have indicated the limitations of our study in the discussion (page 7). Please see the revised manuscript.
Reviewer 2 Report (Previous Reviewer 1)
Page 07, Discussion: Add the name of the countries were swine herds were already for PED before the 2013 outbreaks, since not many used to apply this vaccine.
Page 07, Discussion: Although the authors included the discussion about the levels of IFN in vivo, it is difficult to follow the ideas related to this subject. Consider re-writing the paragraph.
Author Response
Thank you for your comments. On page 7, we listed the names of countries that used PEDV vaccines before 2013.
Following your suggestion, the manuscript has been revised on page 7 and 8.
Please see the revised manuscript.
Reviewer 3 Report (New Reviewer)
The manuscript “Resistance of field-isolated porcine epidemic diarrhea virus to interferon and neutralizing antibody” had the aim is detect factors that are known to be important for PEDV control by comparing the susceptibility against IFN and neutralizing antibodies between G1a CV777-lineage classical and the G2b strains of PEDV. The results will help the understanding of the of the emerging G2b PEDVs strains.
Author Response
Thank you for your review.
This manuscript is a resubmission of an earlier submission. The following is a list of the peer review reports and author responses from that submission.
Round 1
Reviewer 1 Report
- Page 1, first paragraph: "PEDV outbreaks lead to substantial economic losses as the virus is highly contagious (80%–100%) and show significant mortality (50%–90%) in suckling piglets [2]." - The concepts of contagiousness and mortality are not well applied in this sentence. Consider rephrasing.
- Page 2, first paragraph: "We hypothesized that variant PEDVs are..." - include the genogroup and subgroup the variants are classified in, using the same classification as given in the second paragraph of introduction.
- Page 2, "2 Materials and Methods, 2.1. Cells"
- please add the ATCC reference number for the Vero and IPEC-J2 cells, along with details of how they were obtained (specially for IPEC-J2) and the passage they were during the experiments.
- What were the antibiotics used for alveolar macrophages culture and their concentrations?
- Page 2, "2.2. Virus infection": include the word PED before the name of the strains.
- Page 3, "3. Results 3.1. Infectivity of field-isolated PEDV in various cells: - please be consistent with the way the virus strains are named.
- Figures: Legends have to appear below the figure, not above.
- Figure 2, legend: "IPEC-J2 cells were infected with indicated concentra-tions of porcine recombinant IFN-beta for 1 h and then infected with SM98 or PED-CUP-B2014 for 1 h in the presence of 10 μg/ml trypsin and IFN-beta." - Replace "infected" by "treated"
- Page 5: "sera samples" - please correct it for "serum samples"
- Figure 3A and 3B: Use the same scale for both Y axis.
- Discussion, first paragraph: Provide references to support the concerns with PED outbreaks, as the references used are from a decade ago.
- Discussion: According to the results, the addition of trypsin was determinant for the growth of field strain in macrophages and in Vero cells. Please, provide a discussion about the differences of the trypsin treatment for both PEDV strains.
- Discussion: The presented data shows that virus titer was statistically different between isolates only with 250ng/ml of INF-B. How is that relevant in vivo? Is it common to reach this concentration of INF-B in the intestine naturally infected pigs? Include this topic in the discussion.
Reviewer 2 Report
The manuscript does not have the structure of the scientific article, it is more a case report. In case the authors wish to maintain it as an article, they should better address the content and improve the scientific design.
Abstract: Second line: Motility (?)
Introduction:
The introduction must be contextualized, what has been described is not generalizable. Several countries have been managing the disease for decades without major losses, adopting biosecurity and management measures, at the farm level. The virus is highly contagious and causes damage to the “holding”, not to the territory. Veterinary services have not reported the disease to WOAH (World Organization for Animal Health, former OIE) for years, and they should, if the disease had the characteristics, described.
Several countries have managed the disease for decades without major losses. And this should also be explained. It would also be worth investigating whether the inappropriate use of vaccines could have generated the situations described.
The experimental design of the study is not clear as well as the statistical analysis.
The statistical analysis paragraph lacks the description of the methodology for calculating the sample size and the number of samples analysed. The moments in time are not clearly described as well as the statistical techniques used that take time into account.
Furthermore, the table in which the sample is described by group to which it belongs is missing and the software used for the analysis is not declared.